



# 1 NO$_3$ chemistry of wildfire emissions: a kinetic study of the gas-phase
# 2 reactions of furans with the NO$_3$ radical

Mike J. Newland, Yangang Ren, Max R. McGillen, Lisa Michelat, Véronique Daële, Abdelwahid Mellouki
ICARE-CNRS, 1 C Av. de la Recherche Scientifique, 45071 Orléans Cedex 2, France
*Correspondence to*: Mike J. Newland (mike.newland@cnrs-orleans.fr; mellouki@cnrs-orleans.fr)
**Abstract.** Furans are emitted to the atmosphere during biomass burning from the pyrolysis of cellulose. They are one of the
major contributing VOC classes to OH and NO$_3$ reactivity in biomass burning plumes. The major removal process of furans
from the atmosphere at night is reaction with the nitrate radical, NO$_3$. Here we report a series of relative rate experiments in the
7300 L indoor simulation chamber at CNRS-ICARE, Orléans, using a number of different reference compounds to determine
NO$_3$ reaction rate coefficients for four furans, two furanones, and pyrrole. In the case of the two furanones, this is the first time
that NO$_3$ rate coefficients have been reported. The recommended values (cm$^3$ molecule$^{-1}$ s$^{-1}$) are: furan $(1.50\pm0.23)\times10^{-12}$, 2-
methylfuran $(2.37\pm0.55)\times10^{-11}$, 2,5-dimethylfuran $(1.10\pm0.33)\times10^{-10}$, furan-2-aldehyde $(9.28\pm2.3)\times10^{-14}$, 5-methyl-2(3H)-
furanone $(3.00\pm0.45)\times10^{-12}$, 2(5H)-furanone $<1.4\times10^{-16}$, and pyrrole $(7.35\pm2.06)\times10^{-11}$. The furan-2-aldehyde + NO$_3$ reaction
rate is found to be an order of magnitude lower than previously reported. We also recommend a faster rate for the α-
terpinene+NO$_3$ reaction $((2.70\pm0.81)\times10^{-10}$ cm$^3$s$^{-1})$. These experiments show that for furan, alkyl substituted furans, 5-methyl-
2(3H)-furanone, and pyrrole, reaction with NO$_3$ will be the dominant removal process at night, and may also contribute during
the day. For 2(5H)-furanone, reaction with NO$_3$ is not an important atmospheric sink.

## 21 1 Introduction

Furans are five membered aromatic cyclic ethers. Furans (and pyrroles – where N replaces O as the heteroatom) are generated
during the pyrolysis of cellulose and are a major component of emissions from wildfire burning (Hatch et al., 2015, 2017; Koss
et al., 2018; Coggon et al., 2019; Andreae et al., 2019). Such emissions are likely to increase in the future with the spatial extent,
number, and severity, of wildfires globally having increased markedly in recent decades (Jolly et al., 2015; Harvey, 2016) and
predicted to continue to do so as the climate warms (Krikken et al., 2019; Lohmander, 2020). Furans have also been measured
in emissions from residential logwood burning (Hartikainen et al., 2018), and burning of a wide variety of solid-fuels used for
domestic heating and cooking (Stewart et al., 2021a). Furans have been shown to account for a significant proportion of the total
NO$_3$ (Decker et al., 2019) and OH (Koss et al., 2018; Coggon et al., 2019; Stewart et al., 2021b) reactivity of emissions from
burning of typical wildfire and domestic fuels.
Alkyl substituted furans have also been suggested as promising biofuels as they can be derived from lignocellulosic biomass
(Roman-Leshkov et al., 2007; Binder et al., 2009; Wang et al., 2014). This would likely lead to fugitive emissions of these
compounds during distribution, as well as emissions of unburned and partially oxidised products from vehicle exhaust. The
oxidation of certain furan compounds has been shown to have large secondary organic aerosol yields (Hatch et al., 2017;
Hartikainen et al., 2018; Joo et al., 2019, Ahern et al., 2019; Akherati et al., 2020), which could adversely impact air quality.
Oxidation of furans in the atmosphere has been shown to produce 2-furanones (mono-unsaturated five-membered cyclic esters)
both via OH (notably hydroxy-furan-2-ones, Aschmann et al., 2014) and NO$_3$ (Berndt et al., 1997) reactions. 2-Furanones are



also produced from the OH oxidation of six-membered aromatic compounds (Smith et al., 1998, 1999; Hamilton et al., 2005;
Bloss et al., 2005; Wyche et al., 2009; Huang et al., 2015). In both cases, the initial product is thought to be an unsaturated
dicarbonyl, with production of the 2-furanone formed via photoisomerisation of the dicarbonyl to a ketene-enol (Newland et al.,
2019), followed by ring closure of this molecule. In the case of aromatics, the ketene-enol can also be formed directly via
decomposition of the bicyclic peroxy radical intermediate (Wang et al., 2020).
Furan type compounds are removed from the atmosphere by reaction with the major oxidants OH, $NO_3$ and $O_3$. There have been
a number of studies on the rates of reaction of furan type compounds with the dominant daytime oxidant, OH (Lee and Tang,
1982; Atkinson et al., 1983; Wine and Thompson, 1984; Bierbach et al., 1992, 1994, 1995; Aschmann et al., 2011; Ausmeel et
al., 2017; Whelan et al., 2020). However, there have been fewer studies on the rates of reaction of furan type compounds with
the major night-time oxidant, $NO_3$ (Atkinson et al., 1985; Kind et al., 1996; Cabañas et al., 2004; Colmenar et al., 2012).
The nitrate radical, $NO_3$, is produced in the atmosphere predominantly through the reaction of $NO_2$ with $O_3$, and exists in
equilibrium with $N_2O_5$. It has long been known to be an important night-time oxidant (Levy, 1972; Winer et al., 1984). While it
is also produced during the daytime, it is rapidly converted back to $NO_2$ by reaction with NO and by photolysis. However, in
environments with low NO, either due to low NOx emissions, or suppression through high $O_3$ concentrations (e.g. Newland et
al., 2021), $NO_3$ oxidation has been observed to be significant during the day (Hamilton et al., 2021).
Here, we present results of a series of relative rate experiments for furan, 2-methylfuran, 2,5-dimethylfuran, furan-2-aldehyde
(furfural), 5-methyl-2(3H)-furanone (α-angelicalactone), 2(5H)-furanone (γ-crotonolactone), and pyrrole reaction with the $NO_3$
radical, performed in the 7300 L indoor simulation chamber at CNRS-ICARE, Orléans, France.
**2 Experimental**
**2.1    CSA-Chamber**

The CNRS-ICARE indoor chamber is a 7300 L indoor simulation chamber used for studying reaction kinetics and mechanisms
under atmospheric boundary layer conditions. Further details of the chamber setup and instrumentation are available elsewhere
(Zhou et al., 2017). Experiments were performed in the dark at atmospheric pressure (*ca.* 1000 mbar), with the chamber operated
at a slight overpressure to compensate for removal of air for sampling, and to prevent ingress of outside air to the chamber. The
chamber is in a climate controlled room and the temperature was maintained at 299±2 K.

**2.2    Experimental Approach**

Starting with the chamber filled with clean air, the VOCs of interest (*ca.* 3 ppmv) were added, followed by ~ 1 Torr of the inert
gas $SF_6$ to monitor the chamber dilution rate. The chamber was left for at least thirty minutes prior to the start of the experiment
to monitor the dilution rate and losses of the VOCs to the chamber walls. These losses, $(1 - 8) \times 10^{-6}\,s^{-1}$, were always smaller than
dilution (~$1.2 \times 10^{-5}\,s^{-1}$). The reaction was then initiated by continually introducing an $N_2O_5$ sample, held in a trap at ~ 235 K,
with air flow of $(2.5 - 5)$ L/min through it. The chamber was monitored until most of the VOC of interest was consumed, with
experiments generally taking 0.5 – 2 hours. The experiments were performed under dry conditions (RH ≤ 1.5 %).


VOC abundance was determined by *in-situ* Fourier Transform Infrared (FTIR) Spectroscopy using a Nicolet 5700 coupled to a
White-type multipass cell with a pathlength of 143 m. Each scan was comprised of either 30 or 60 co-additions depending on
the expected rate of loss of the VOCs.

**2.3      Materials**

Furan (>99%, Sigma-Aldrich), 2-methylfuran (>98%, TCI), 2,5-dimethylfuran (>98%, TCI), pyrrole (>99%, TCI), α-
angelicalactone (>98%, TCI), furfural (>98%, TCI), α-terpinene (90%, Sigma-Aldrich), 2,3-dimethyl-but-2-ene (98%, Sigma-
Aldrich), 2-carene (97%, Sigma-Aldrich), camphene (95%, Sigma-Aldrich), α-pinene (98%, Sigma-Aldrich), cyclohexene
(≥99%, Sigma-Aldrich), 3-methyl-3-buten-1-ol (97%, Sigma-Aldrich) and γ-crotonolactone (>93%, TCI), were used as supplied
without further purification.
$N_2O_5$ was synthesised by reacting $NO_2$ with excess $O_3$. First, $NO$ and $O_3$ were mixed to generate $NO_2$ (Reaction R1). This $NO_2$ /
$O_3$ mixture was then flushed into a bulb in which $NO_3$ and subsequently $N_2O_5$ were generated through Reactions R2-R3.

$NO + O_3 \rightarrow NO_2$        (R1)
$NO_2 + O_3 \rightarrow NO_3$       (R2)
$NO_2 + NO_3 \rightarrow N_2O_5$     (R3)

$N_2O_5$ crystals were then collected in a cold trap at 190K. The $N_2O_5$ sample was purified by trap to trap distillation under a flow
of $O_2$ / $O_3$. The final sample was stored at 190 K and used within a week.

**2.4      Analysis**

VOC concentrations were monitored by FTIR. The furans generally have a number of major absorption bands in the infrared.
The main bands used for analysis are shown in Table 1 (bold), as well as other characteristic bands for each compound. Reference
spectra of the major bands for each compound taken in the chamber at a resolution of 0.25 cm$^{-1}$ are provided in the Supplement
(Figures S1-S7). The ANIR curve fitting software (Ródenas, 2018), which implements a least squares fitting algorithm was used
to generate time profiles for each compound based on their reference spectra. Profiles were checked by doing a number of manual
subtractions. Example time profiles from an experiment with α-angelicalactone and furan, with cyclohexene as the reference
compound, are shown in Figure 1. Relative rate plots from the experiments with furan and 2-methylfuran are shown in Figure 2.









**Table 1** Maxima of major absorption bands (of Q branches if present) for the compounds used in this study. Bands used
predominantly for analysis are shown in bold.

| Compound | Main absorption bands / cm⁻¹ |
|---|---|
| Furan | **995**, 744 |
| 2-Methylfuran | **792**, 726, 1151, 2965 |
| 2,5-Dimethylfuran | **777**, 2938, 2961 |
| Furfural | **756**, 1720 |
| Pyrrole | **724**, 1017, 3531, 718-722 |
| 5-Methyl-2(3H)-furanone | **731, 939**, 1100, 1834 |
| 2(5H)-furanone | **1098,** 805, 866, 1045, 1812, 2885, 2945 |


Relative rate experiments were performed, whereby a compound (or two) with an unknown reaction rate ($k_{VOC}$) with $NO_3$ was
added to the chamber with a reference compound with a known $NO_3$ reaction rate ($k_{ref}$). A plot of the relative loss of the compound
against the reference compound following addition of $NO_3$ (via $N_2O_5$ decomposition), accounting for both chamber dilution and
wall losses ($k_d$), gives a gradient of $k_{VOC}/k_{ref}$ (Equation E1).

$$\ln\frac{([VOC]_0)}{([VOC]_t)} - k_d t = \frac{k_{VOC}}{k_{ref}}\ln\frac{[ref]_0}{[ref]_t} - k_d t \qquad (E1)$$

A number of reference compounds were used for each VOC, chosen so that the reference rate was roughly within a factor of five
of the expected unknown rate, and with an attempt to use different references that had both faster and slower $NO_3$ reaction rates
than the VOC. Reaction rates of the reference compounds (Table 2) are taken from the Database for the Kinetics of the Gas-
Phase Atmospheric Reactions of Organic Compounds v2.1.0 (McGillen et al., 2020), available at data.eurochamp.org/data-
access/kin/#/home.

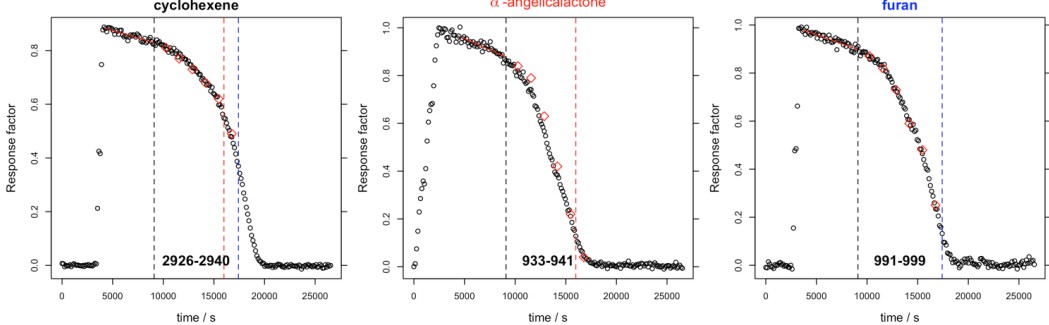


**Figure 1** Concentration-time profiles from experiment with cyclohexene, α-angelicalactone and furan. Black circles are response
factors generated by the ANIR curve fitting program relative to the reference spectra. Red diamonds are obtained from manual
subtractions. Left black dashed vertical line is the beginning of the region used for the relative rate calculation, the red dashed
line is the end of the region used for the calculation of the α-angelicalactone relative rate, the blue line is the end of the region
used for the calculation of the furan relative rate. Bold values at the bottom are the absorption bands used for analysis.





**Table 2.** Reference compounds used. Recommended rates and uncertainties from McGillen et al. (2020).

| Compound | $k$ / cm$^3$ molecule$^{-1}$ s$^{-1}$ |
|---|---|
| 2,3-dimethyl-2-butene | $(5.70\pm1.71)\times10^{-11}$ |
| 2-carene | $(2.0\pm0.3)\times10^{-11}$ |
| $\alpha$-pinene | $(6.20\pm1.55)\times10^{-12}$ |
| camphene | $(6.60\pm1.65)\times10^{-13}$ |
| cyclohexene | $(5.60\pm0.84)\times10^{-13}$ |
| 3-methyl-3-buten-1-ol | $(2.60\pm0.78)\times10^{-13}$ |


It is noted that no OH scavenger was used in these experiments (as is the case for most, if not all, NO$_3$ previous relative rate
studies to the authors' knowledge). NO$_3$ reaction with alkenes tends to proceed by electrophilic addition to the double bond
followed by addition of O$_2$ to the resulting radical, leading to a nitrooxy peroxy radical ($\beta$-ONO$_2$-RO$_2$) (Barnes et al., 1989;
Hjorth et al., 1990). It has recently been shown (Novelli et al., 2021) that there is the possibility of OH formation through the
reactions of $\beta$-ONO$_2$-RO$_2$ with HO$_2$. HO$_2$ could be generated in these experiments from the abstraction of an H atom by O$_2$ from
a $\beta$-ONO$_2$-RO radical with available H atoms. The initial NO$_3$ reaction with furans is not thought to form $\beta$-ONO$_2$-RO$_2$ radicals,
with NO$_3$ addition to the C2 carbon followed by O$_2$ addition to the C5 carbon (Berndt et al., 1996), analogous to the OH addition
reaction (Bierbach et al., 1995; Mousavipour et al., 2009; Yuan et al., 2017; Whelan et al., 2020). However, some of the reference
compounds used in the experiments will form such radicals. For example, the reaction of HO$_2$ with the $\beta$-ONO$_2$-RO$_2$ radicals
formed from $\alpha$-pinene + NO$_3$ has been reported to have an OH yield of up to 70 % (Kurtén et al., 2017). A box model run was
performed to test the impact of this chemistry in this study. The $\alpha$-pinene scheme from the MCMv3.3.1 (Jenkin et al., 1997;
mcm.york.ac.uk) was incorporated into the box model AtChem (Sommariva et al., 2020), and an OH yield of 0.5 was assigned
to the reaction of HO$_2$ with the initial $\beta$-ONO$_2$-RO$_2$ radicals formed from the $\alpha$-pinene+NO$_3$ reaction. The model was initiated
with 2-methylfuran and $\alpha$-pinene concentrations of 3 ppmv, representative of the experiments performed here. NO$_3$
concentrations were constrained to give a lifetime of ~ 1 hour for the VOCs, typical of the experiments. OH reaction was found
to account for less than 1 % of the removal of 2-methylfuran or $\alpha$-pinene through the model run. Consequently, it can be assumed
that OH chemistry is a negligible interference in these experiments.



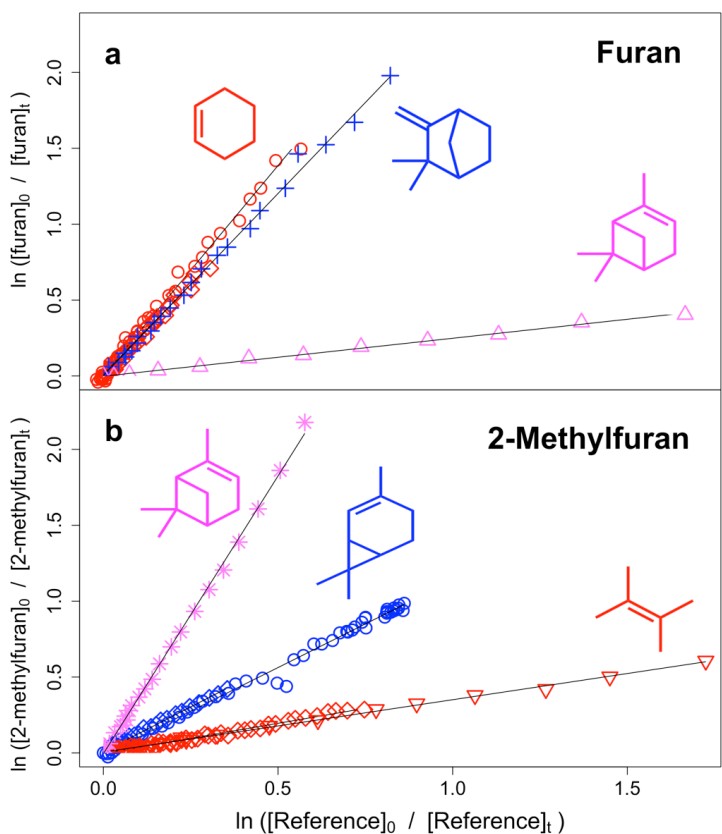


**Figure 2.** Relative rate plots for: **a.** furan relative to cyclohexene (red), camphene (blue), and α-pinene (pink); **b.** 2-methylfuran

relative to 2-carene (blue), 2,3-dimethyl-2-butene (red), and α-pinene (pink). Different shapes are used for different experiments

with the same reference compound.

**3 Results and Discussion**
The $k(NO_3)$ rate coefficients determined with each reference compound are given in Table 3 and Figure 3. A recommendation
of an updated rate coefficient for α-terpinene+$NO_3$ is also given in Table 3. Overall recommended values for the rate coefficient
for each compound are calculated by taking the mean (weighted by the reported uncertainty of the reference) of the rate
coefficient derived from each experiment with each reference compound, including using the recommended values for the other
furans and for α-terpinene presented in Table 3. Uncertainties for the relative rates in Table S1 are assumed to be < 10 % and to
be dominated by statistical errors in fitting to the absorption bands. Uncertainties for the rate coefficients reported in Table 3 are
dominated by the assumed uncertainties in $k(NO_3)$ of the reference compounds. For most of the references, the uncertainties are
20 – 30 %, taken from the recommendations of McGillen et al. (2020). For 2,3-dimethyl-2-butene, the recommended uncertainty



in McGillen et al. (2020) is 150 %, but based on the fact that the rate coefficients derived using 2,3-dimethyl-2-butene for 2-
methylfuran, 2,5-dimethylfuran and pyrrole agree very well with those using other references with much smaller uncertainties,
a conservative estimate of 30 % is used here. It is noted that for all compounds, the rate coefficients derived with different
references agree very well, to within 10%, with the exception of α-terpinene, which is discussed further below. The
experimentally determined $k(NO_3)$ rates of the furans relative to each other are in good agreement (to within 6%) with those
calculated using the weighted means shown in Table 3 (Table S2). This gives further confidence in the $k(NO_3)$ values used for
the reference compounds.

The rate coefficient derived for furan, agrees well with the value previously reported by Atkinson et al. (1985) from a chamber
relative rate experiment. However, there is significant differences between the values reported here for furan, 2-methylfuran and
2,5-dimethylfuran, and those reported by Kind et al. (1996) from relative rate experiments in a flow reactor. While the value
reported for 2-methylfuran agrees within the uncertainties between the two studies, the values for furan and 2,5-dimethylfuran
are ~ 50 % and 100 % greater respectively. It is unclear what is behind this observed disparity; the good agreement between the
two studies for the 2-methylfuran rate coefficient suggests that there is not a systematic difference between the experimental
setups. For pyrrole, the rate coefficient determined here is about 50% faster than the value reported by Atkinson et al. (1985)
from a chamber relative rate experiment using $N_2O_5$ thermal decomposition. Cabañas et al. (2004) reported an upper limit of
$<1.8\times10^{-10}$ cm$^3$ molecule$^{-1}$ s$^{-1}$ (298K) using an absolute technique of fast flow discharge.
For 2-furanaldehyde (furfural) + $NO_3$, the rate coefficient recommended here is an order of magnitude slower than the only
previously reported values (Colmenar et al., 2012), derived from small chamber relative rate experiments with 2-methyl-2-butene
and α-pinene as references. The rate coefficient from Colmenar et al. (2012) is very similar to the reported rate coefficient for
furan+$NO_3$. This is surprising, since the presence of a formyl group attached to a double bond is expected to be strongly
deactivating with respect to addition to that bond, due to the electron withdrawing mesomeric effect of the –C(O)H group
(Kerdouci et al., 2014). This has also been observed for other electrophilic addition reactions, such as those with OH and $O_3$
(Kwok and Atkinson, 1995; McGillen et al, 2011; Jenkin et al., 2020). And while there is the possibility of H abstraction from
the formyl group, which would increase the overall rate coefficient, such reactions are typically of the order of $10^{-14}$ cm$^3$s$^{-1}$
(Kerdouci et al., 2014), and hence would not be expected to compensate for the reduced rate of the addition reaction.
For 5-methyl-(3H)-furan-2-one (α-angelica lactone) + $NO_3$ this is the first reported rate coefficient. For (5H)-furan-2-one (γ-
crotonolactone), relative rate experiments with several reference compounds were attempted, with the slowest reacting of these
being cyclohexane ($k_{NO3}$ = $1.4\times10^{-16}$ cm$^3$ molecule$^{-1}$ s$^{-1}$). Roughly 10 % of the cyclohexane was removed in this experiment
(accounting for loss by dilution), with no appreciable loss of γ-crotonolactone. We can therefore deduce that $k$(γ-
crotonolactone+$NO_3$) << $1.4\times10^{-16}$ cm$^3$ molecule$^{-1}$ s$^{-1}$. Again, this is the first time a $NO_3$ reaction rate has been measured for this
compound. A comparison of the two furanones shows that 5-methyl-(3H)-furan-2-one reacts more than four orders of magnitude
faster than (5H)-furan-2-one. This can be explained in part by the presence of a methyl group, which is seen to increase the rate
by roughly an order of magnitude from e.g. furan to 2-methylfuran to 2,5-dimethylfuran. Berndt et al. (1997) derived an $NO_3$
reaction rate of $1.76\times10^{-13}$ cm$^3$ molecule$^{-1}$ s$^{-1}$ for (3H)-furan-2-one. However, the majority of the difference must be explained by
the structure of the two compounds, namely the conjugated nature of the C=C and C=O bonds in (5H)-furan-2-one. The carbonyl
group removes electron density from the C=C bond greatly reducing the rate coefficient. A similar relationship is seen for



analogous acyclic compounds e.g. the $NO_3$ rate coefficient of the conjugated ester methyl acrylate is almost two orders of
magnitude greater than that of the non-conjugated isomer vinyl acetate.


**Table 3.** $NO_3$ reaction rate coefficients derived for each experiment and recommended value based on the weighted mean.

| Compound | Reference (repeats) | $k(NO_3)$ / $cm^3$ molecule$^{-1}$ s$^{-1}$ | Weighted mean $k(NO_3)$ / $cm^3$ molecule$^{-1}$ s$^{-1}$ |
|---|---|---|---|
| *α-terpinene* | 2,5-dimethylfuran (1) | $2.92 \times 10^{-10}$ | $\mathbf{2.74 \pm 0.81 \times 10^{-10}}$ |
| | 2,3-dimethyl-2-butene (1) | $2.50 \times 10^{-10}$ | |
| | pyrrole (1) | $2.69 \times 10^{-10}$ | |
| *2,5-dimethylfuran* | 2-carene (1) | $1.12 \times 10^{-10}$ | $\mathbf{1.10 \pm 0.33 \times 10^{-10}}$ |
| | 2,3-dimethyl-2-butene (1) | $1.21 \times 10^{-10}$ | |
| | a-terpinene (1) | $1.01 \times 10^{-10}$ | |
| | pyrrole (1) | $1.10 \times 10^{-10}$ | |
| | 2-methylfuran (2) | $1.07 \times 10^{-10}$ | |
| *pyrrole* | 2-carene (1) | $7.68 \times 10^{-11}$ | $\mathbf{7.35 \pm 2.06 \times 10^{-11}}$ |
| | 2,3-dimethyl-2-butene (2) | $6.87 \times 10^{-11}$ | |
| | a-terpinene (1) | $7.15 \times 10^{-11}$ | |
| | 2,5-dimethylfuran (1) | $7.22 \times 10^{-11}$ | |
| | 2-methylfuran (2) | $7.52 \times 10^{-11}$ | |
| *2-methylfuran* | 2-carene (3) | $2.47 \times 10^{-11}$ | $\mathbf{2.37 \pm 0.55 \times 10^{-11}}$ |
| | 2,3-dimethyl-2-butene (2) | $2.12 \times 10^{-11}$ | |
| | α-pinene (1) | $2.27 \times 10^{-11}$ | |
| | pyrrole (2) | $2.28 \times 10^{-11}$ | |
| | 2,5-dimethylfuran (2) | $2.41 \times 10^{-11}$ | |
| *α-angelicalactone* | α-pinene | $3.03 \times 10^{-12}$ | $\mathbf{3.00 \pm 0.45 \times 10^{-12}}$ |
| | cyclohexene | $2.89 \times 10^{-12}$ | |
| | furan (2) | $3.05 \times 10^{-12}$ | |
| *furan* | cyclohexene | $1.46 \times 10^{-12}$ | $\mathbf{1.50 \pm 0.23 \times 10^{-12}}$ |
| | α-pinene | $1.55 \times 10^{-12}$ | |
| | camphene | $1.58 \times 10^{-12}$ | |
| | α-angelicalactone (2) | $1.49 \times 10^{-12}$ | |
| *furfural* | cyclohexene (1) | $9.02 \times 10^{-14}$ | $\mathbf{9.28 \pm 2.30 \times 10^{-14}}$ |
| | 3-methyl-3-buten-1-ol (1) | $9.54 \times 10^{-14}$ | |
| | camphene (1) | $9.50 \times 10^{-14}$ | |
| *γ-crotonolactone* | cyclohexane | $< 1.4 \times 10^{-16}$ | $\mathbf{< 1.4 \times 10^{-16}}$ |



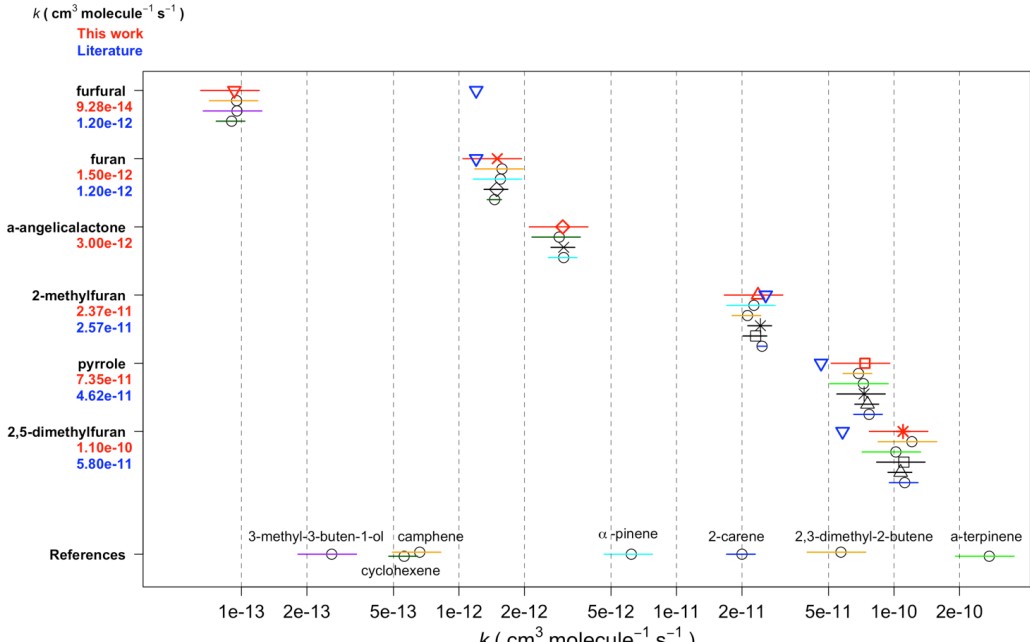


**Figure 3** The reaction rate coefficients derived for the six compounds in this work (excluding α-terpinene). Red triangles (and
red text, left axis) represent the weighted mean of all experiments in this work, blue inverted triangles (and blue text, left axis)
are the recommended values from McGillen et al. (2020). Horizontal lines represent uncertainty in rate coefficient, colours
(shapes if other furans) represent which reference was used.

α-terpinene was used as a reference compound in two experiments. However, the rate coefficients derived for 2,5-dimethylfuran
and pyrrole are significantly smaller using α-terpinene compared to other reference compounds. In addition, its reaction rate
relative to 2,3-dimethyl-2-butene is larger than expected based on the recommended rate coefficient of $(1.80\pm1.44)\times10^{-10}$ cm$^3$
molecule$^{-1}$ s$^{-1}$ (McGillen et al., 2020). The reaction with α-terpinene is one of the largest known VOC+NO$_3$ rate coefficients,
and hence it is a useful reference and important to know the rate with a good degree of certainty. We derive a rate coefficient
relative to 2,5-dimethylfuran of 2.68, to pyrrole of 3.79 and to 2,3-dimethyl-2-butene of 4.39. Using the recommended values
given in Table 3 for 2,5-dimethylfuran and pyrrole, and the recommended value for 2,3-dimethyl-2-butene in Table 2, gives an
average NO$_3$ reaction rate for α-terpinene of $(2.70\pm0.81)\times10^{-10}$ cm$^3$ molecule$^{-1}$ s$^{-1}$. This is considerably faster than a recent
absolute rate measurement of $(1.2\pm0.3)\times10^{-10}$ cm$^3$ molecule$^{-1}$ s$^{-1}$ (Fouqueau et al., 2020), and previous relative rate determinations
of $(1.6\pm0.6)\times10^{-10}$ cm$^3$ molecule$^{-1}$ s$^{-1}$ (Atkinson et al., 1985) and $(0.9\pm0.4)\times10^{-10}$ (Berndt et al., 1996) using TME as a reference.






**Table 4.** Recommended $NO_3$ rate coefficients from this work compared to those reported in the literature.

| Compound | Rate / cm³ molecule⁻¹ s⁻¹ | Reference | Technique | NO₃ source |
|---|---|---|---|---|
| *α-terpinene* | **(2.74±0.81)×10⁻¹⁰** | **This work** | | |
| | (1.6±0.6)×10⁻¹⁰ | Atkinson et al. (1985) | Relative (2,3-dimethyl-2-butene) | |
| | (0.9±0.4)×10⁻¹⁰ | Berndt et al. (1996) | Relative (2,3-dimethyl-2-butene) | |
| | (1.2±0.3)×10⁻¹⁰ | Fouqueau et al. (2020) | | |
| *2,5-dimethylfuran* | **(1.10±0.33)×10⁻¹⁰** | **This work** | | |
| | (5.78±0.34)×10⁻¹¹ | Kind et al. (2006) | Flow reactor: relative (*trans*-2-butene) | N₂O₅ |
| *pyrrole* | **(7.35±2.06)×10⁻¹¹** | **This work** | | |
| | (4.6±1.1)×10⁻¹¹ | Atkinson et al. (1985) | Chamber: relative (2-methyl-2-butene) | N₂O₅ |
| | < 1×10⁻¹⁰ | Cabañas et al. (2004) | Flow reactor: absolute (LIF detection of NO₃) | HNO₃+F |
| *2-methylfuran* | **(2.37±0.55)×10⁻¹¹** | **This work** | | |
| | (2.57±0.17)×10⁻¹¹ | Kind et al. (2006) | Flow reactor: relative (*trans*-2-butene) | N₂O₅ |
| *α-angelicalactone* | **(3.00±0.45)×10⁻¹²** | **This work** | | |
| *furan* | **(1.50±0.23)×10⁻¹²** | **This work** | | |
| | (1.5±0.2)×10⁻¹² ᵃ | Atkinson et al. (1985) | Chamber: relative (*trans*-2-butene) | N₂O₅ |
| | (0.998±0.062)×10⁻¹² | Kind et al. (2006) | Flow reactor: relative (*trans*-2-butene) | N₂O₅ |
| | (1.36±0.8)×10⁻¹² | Cabañas et al. (2004) | Flow reactor: absolute (LIF detection of NO₃) | HNO₃+F |
| *furfural* | **(9.28±2.30)×10⁻¹⁴** | **This work** | | |
| | (1.17±0.15)×10⁻¹² | Colmenar et al. (2012) | Small chamber: relative (2-methyl-2-butene) | N₂O₅ |
| | (1.36±0.38)×10⁻¹² | Colmenar et al. (2012) | Small chamber: relative (α-pinene) | N₂O₅ |
| *γ-crotonolactone* | **< 1.4×10⁻¹⁶** | **This work** | | |

ᵃ corrected for change to recommended rate for reference (trans-2-butene)

**Atmospheric implications**

The atmospheric lifetimes of the compounds, based on the rate coefficients reported herein, are given in Table 5. These assume
concentrations of OH = $5×10^6$ molecules cm⁻³ (typical daily peak summertime concentrations $1.5×10^6$ – $1.5×10^7$ molecules cm⁻
³ (Stone et al., 2012)), night-time $NO_3$ = $2×10^8$ molecules cm⁻³ (typical night-time concentrations $1×10^8$ – >$1×10^9$ cm⁻³ (Brown
and Stutz (2012)) daytime $NO_3$ = $1×10^7$ molecules cm⁻³ (limited daytime measurements suggest concentrations ~ 0.5 – >1 pptv
($2.5×10^7$ molecules cm⁻³) (Brown and Stutz (2012)), and $O_3$ = 40 ppbv (background $O_3$ concentration ~ 40 ppb (Parrish et al.,
2014)). From these values it is clear that the alkyl substituted furans and pyrrole have very short lifetimes both during the day,
when the dominant daytime sink is likely to be reaction with OH, and at night, when the dominant sink will be reaction with
$NO_3$. $O_3$ may contribute somewhat to the removal of these compounds both during the day and night, particularly for 2,5-
dimethylfuran. As $k(NO_3)$ approaches the same order of magnitude as $k(OH)$, e.g. for 2-methylfuran, 2,5-dimethylfuran and
pyrrole, the $NO_3$ reaction is likely to be competitive with the OH reaction even during the day in low NOx environments, with



daytime NO$_3$ concentrations reported to be ~ 1 ppt (2.5×10$^7$ molecules cm$^{-3}$) (Brown and Stutz, 2012). The relatively large rate
coefficient reported here for 5-methyl-2(3H)-furanone, suggests that NO$_3$ reaction will be an important sink for unsaturated non-
conjugated cyclic esters. On the other hand, the very slow rate of the 2(5H)-furanone+NO$_3$ reaction suggests that this will not be
an important atmospheric sink. 2(5H)-furanone has also been shown to have a very slow reaction with O$_3$ (lifetime > 100 years,
Ausmeel et al., 2017), whereas for reaction with OH, the lifetime is much shorter, and this will be the predominant gas-phase
sink for 2(5H)-furanone. Such a slow NO$_3$ reaction might be expected to extend to all 2-furanones with a conjugated structure,
e.g. hydroxyfuranones – major products of OH oxidation of methyl substituted furans (Aschmann et al., 2014), such that the
nitrate reaction may be unimportant in the atmosphere for these structures. Although substitution at the double bond is likely to
increase the rate coefficient somewhat, as observed for OH and O$_3$ reactions with the methyl-substituted form of 2(5H)-furanone
(Ausmeel et al., 2017).
One of the major sources of furan type compounds to the atmosphere is wildfires. Wildfire plumes can be regions of high NO$_3$
even during the day due to suppressed photolysis rates in optically thick plumes (Decker et al, 2021). NO$_3$ oxidation of furans
may be even more important in such plumes than in the background atmosphere. Such plumes can extend over hundreds of
kilometres and hence affect air quality on a local and regional scale (e.g. Andreae et al., 1988; Brocchi et al., 2018; Johnson et
al., 2021). Domestic wood burning is an increasing trend in northern European cities (Chafe et al., 2015). Burning will generally
be in the winter during which, with short daylight hours and peak daytime OH often an order of magnitude lower than during
the summer, the reaction with NO$_3$ is likely to be the dominant fate of furan type compounds in such emissions, contributing
significantly to organic aerosol in urban areas (Kodros et al., 2020).
Berndt et al. (1996) identified the major first generation products of furan+NO$_3$ to be the unsaturated dicarbonyl, butenedial, and
2(3H)-furanone, with the NO$_3$ recycled back to NO$_2$. However, Tapia et al. (2011), and Joo et al. (2019) found that the major
products of the 3-methylfuran+NO$_3$ reaction predominantly retain the NO$_3$ functionality. In this case, furan+NO$_3$ oxidation
chemistry may be a significant sink for NOx, sequestering it in nitrate species, which might release it far from source on further
gas-phase oxidation, or, due to their low volatility, be taken up into aerosol (Joo et al. 2019).















**Table 5.** Atmospheric gas-phase lifetimes of the compounds reported herein based on typical mid-day OH concentrations of $5\times10^6$ molecules cm$^{-3}$, night-time NO$_3$ concentrations of $2\times10^8$ molecules cm$^{-3}$, day-time NO$_3$ concentrations of $1\times10^7$ molecules cm$^{-3}$, and background O$_3$ concentrations of 40 ppbv ($1\times10^{12}$ molecules cm$^{-3}$).

| Compound | $\tau_{NO3\ (night)}$ | $\tau_{NO3\ (day)}$ | $\tau_{OH\ (day)}$ | $\tau_{O3}$ | $\tau_{total\ (day)}$ |
|---|---|---|---|---|---|
| 2,5-dimethylfuran | 0.76 min | 15 min | 26 min [a] | 40 min [b] | 8 min |
| 2-methylfuran | 3.5 min | 70 min | 48 min [a] | - | 28 min |
| furan | 56 min | 19 hours | 83 min [a] | 116 hours [c] | 77 min |
| pyrrole | 1.1 min | 23 min | 28 min [d] | 18 hours [d] | 13 mins |
| furfural | 15 hours | 12 days | 95 min [e] | - | 94 min |
| 5-methyl-2(3H)-furanone | 28 min | 9.3 hours | 48 min [f] | 3.5 hours [g] | 37 mins |
| 2(5H)-furanone | > 1.1 year | > 22 years | 14 hours [h] | 173 years | 14 hours |

[a] Matsumoto (2011); [b] Dillon et al. (2012); [c] Atkinson et al. (1983); [d] Atkinson et al. (1984); [e] Bierbach et al. (1995); [f] Bierbach et al. (1994); [g] estimated (Bierbach et al., 1994); [h] Ausmeel et al. (2017)

281

**4 Conclusions**

Rate coefficients are recommended for reaction of seven furan type VOCs with NO$_3$ at 298 K and 760 Torr, based on a series of relative rate experiments. These new recommendations highlight the importance of NO$_3$ chemistry to the removal of furans, and other similar VOCs, under atmospheric conditions. The measured rate coefficients suggest that for the three furans reported here, as well as for pyrrole and 5-methyl-2(3H)-furanone, reaction with NO$_3$ is likely to be their dominant night-time sink. For the alkyl furans and pyrrole, reaction with NO$_3$ may also be a significant sink during the daytime. In addition to the rates for the furan type compounds, an updated recommendation is provided for $k(\alpha\text{-terpinene}+NO_3)$, a reaction of particular importance as one of the fastest known VOC+NO$_3$ reaction rate coefficients. This work also extends the existing database of VOC+NO$_3$ reactions, providing valuable reference values for future work.

*Data availability.* All relevant data and supporting information have been provided in the Supplement.





*Author contributions.* MJN performed the experiments with the technical support of YR and MRM and performed the data
treatment and interpretation. MJN wrote the paper. All coauthors revised the content of the original manuscript and approved
the final version of the paper.

*Competing interests.*
The authors declare that they have no conflict of interest.

*Special issue statement.* This article is part of the special issue "Simulation chambers as tools in atmospheric research
(AMT/ACP/GMD inter-journal SI)". It is not associated with a conference.

*Acknowledgements.*
This work is supported by the European Union's Horizon 2020 research and innovation program through the EUROCHAMP-
2020 Infrastructure Activity under grant agreement no. 730997, Labex Voltaire (ANR-10-LABX-100-01) and ANR (SEA_M
project, ANR-16-CE01-0013, program ANR-RGC 2016).

Financial support. This research has been supported by the European Commission Horizon 2020 Framework Programme (grant
no. EUROCHAMP-2020 (730997)) and the Agence Nationale de la Recherche (grants nos. ANR-10-LABX-100-01 and ANR-
16-CE01-0013)

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
