# Peer review of "NO3 chemistry of wildfire emissions: a kinetic study of the gas-phase reactions of furans with the NO3 radical"

_Atmospheric Chemistry and Physics, 2021_

## Author Comment (AC1)

**Response to Reviewers**

**23/11/21**

**Response to Reviewers of:**

**$NO_3$ chemistry of wildfire emissions: a kinetic study of the gas-phase reactions of furans with the $NO_3$ radical by Newland et al., 2021, submitted to ACP**

**General Response**

We thank the reviewers for giving up their time to make insightful comments, helping to clarify and improve our manuscript. The reviewers recognise the importance of the results presented, and recommend publication in ACP after some minor changes. All changes to the manuscript are in line with the reviewers' comments and suggestions.

Responses to each reviewer are given below. Responses to specific points raised by each reviewer are given separately beneath that point. Reviewers' comments are bold and italic, the authors' comments are inset in plain type.

**Major changes**

Many of the reviewers' concerns are focused on the experiments using $\alpha$-terpinene as a reference compound, and the recommendation of a $NO_3$ rate coefficient larger than the three previous determinations (which we note are not in agreement). Reviewer 1 raised the possibility that we had overlooked the potential for the $\alpha$-terpinene+$NO_2$ reaction to impact our results under the experimental conditions employed. This was correct. Having performed experiments with $\alpha$-terpinene+$NO_2$, we obtained a rate coefficient in agreement with that previously determined by Atkinson et al. (1984). Further details of the $NO_2$ experiments with $\alpha$-terpinene and several of the furans are given in response to specific comments below. However, using this value for $k(\alpha$-terpinene+$NO_2$), with the measured $NO_2$ (FTIR), to correct the relative rate calculations led to disagreement between our experimental results. The experiment with $\alpha$-terpinene and 2,5-dimethylfuran gave a rate coefficient relative to $k(2,5$-dimethylfuran+$NO_3$) of about 2.3, giving a rate coefficient for $k(\alpha$-terpinene+$NO_3$) of about $2.5 \times 10^{-10}$ $cm^3$ molecule$^{-1}$ s$^{-1}$, based on our determination of $k(2,5$-dimethylfuran+$NO_3$). Whereas, the experiment using 2,3-dimethyl-2-butene as a reference compound, after correction gives a rate relative to 2,3-dimethyl-2-butene of about 2.8, yielding a value for $k(\alpha$-terpinene+ $NO_3$) of $1.6 \times 10^{-10}$ $cm^3$ molecule$^{-1}$ s$^{-1}$, in line with that reported by Atkinson et al. (1985).

In light of the disagreement between our results and the reported values, and since the manuscript is not explicitly about $\alpha$-terpinene, it was merely one of several reference compounds, it seems prudent to remove the experiments using $\alpha$-terpinene as a reference from the manuscript. We will continue to explore the system, which is clearly important atmospherically, interesting from a chemistry standpoint, and important as a reference compound with one of the largest $NO_3$ reaction rate coefficients known.

The recommended 2,5-dimethylfuran and pyrrole rate coefficients are still tied to at least three reference compounds.

It should be noted that the removal of the α-terpinene experiments has reduced the recommended rate coefficients of the furans slightly.

60

**Data Presentation / Availability**

Figure 2 is now a six panel plot containing relative rate plots for all experiments for all six compounds (furan, 2-methylfuran, 2,5-dimethylfuran, furfural, pyrrole, α-angelicalactone).

65

The SI now contains an additional six concentration-time plots, with the same setup as Figure 1.

All of the data (raw output from the FTIR, and concentration-time data) will be made available at https://data.eurochamp.org with a doi.

70

**Anonymous Referee #1**

*A relative rate study is reported, which considers the reactions of NO₃ with a series of furans and related*
75  *compounds, which are known to be important components of biomass burning emissions. The experiments were carried out in the large CNRS-ICARE chamber, using N₂O₅ decomposition as the source of NO₃, with the rate coefficients being mainly determined relative to those of a series of alkenes of comparable reactivities. The target compounds include furan, 2-methylfuran, 2,5-dimethylfuran, furan-2-aldehyde, 5-methyl-2(3H)-furanone, 2(5H)-furanone and pyrrole. In the case of 5-methyl-2(3H)-furanone, this is the first reported*
80  *determination. A rate coefficient for the reaction of NO₃ with the reactive monoterpene, alpha-terpinene, is also reported which is higher than previous absolute and relative rate determinations.*

*This paper considers an important topic, where new and confirmatory kinetic data are required, and the paper is clearly presented and written. I have one major comment on the experimental set-up and its interpretation,*
85  *and some minor comments, which are outlined below. The major comment relates to the possible impact of NO₂ chemistry in the system, particularly for the α-terpinene rate coefficient determination, but which should probably also be checked in other cases where the current and previous results differ. I have submitted this review promptly, and I hope this will give the authors time to consider this and make adjustments, if required.*

90

*Major Comment*

*The NO₃ source employed (N₂O₅ decomposition) produces an equivalent amount of NO₂, and additional NO₂ is likely formed as a product of the NO₃-initiated organic chemistry (i.e., from decomposition of nitro-oxy oxy*
95  *radicals). Given the timescale of the experiments (0.5 – 2 hours), and the lack of reaction partners for NO₂, the potential reaction of NO₂ with the unsaturated organics needs to be considered and assessed, to confirm that the assumption of NO₃ being the only reagent is valid (i.e., Eq. (E1)).*

*NO₂ has previously been shown to react only very slowly with monoalkenes, but systematically more rapidly*
100  *with conjugated dienes – and with further increases in reactivity resulting from alkyl substitution and in cyclohexadiene rings (Atkinson et al., 1984; Niki et al., 1986; Ohta et al., 1986; Jenkin et al., 2005; Bernard et al., 2013). The rate coefficient reported for α-terpinene (1-isopropyl-4-methyl-cyclohexa-1,3-diene: $6.5 \times 10^{-18}$ cm³ molecule⁻¹ s⁻¹) is therefore one of the highest measured to date (Atkinson et al., 1984). The lifetime of NO₂ with respect to reaction with 3 ppm alpha-terpinene is therefore about 0.5 hours, such that some*
105  *supplementary removal of α-terpinene by this reaction is likely to have occurred under the conditions employed in the present work. In contrast, the alkene reference compound, 2,3-dimethyl-2-butene, reacts much more slowly with NO₂ ($1.5 \times 10^{-20}$ cm³ molecule⁻¹ s⁻¹). As a result, this interference may contribute to the present k(NO₃) determination for α-terpinene being higher than all previous determinations. The previous relative rate studies either corrected for NO₂ reaction (Atkinson et al., 1985) or employed flowing systems with short residence times*
110  *(Berndt et al., 1996). The authors therefore need to check for this potential interference (which presumably would be a very straightforward experiment) and make corrections if required.*

*For completeness, the possible reaction of $NO_2$ with the furans etc. also ideally needs to be checked, although these are likely slower reactions than for α-terpinene. Atkinson et al. (1985) verified that the $NO_2$ reaction was unimportant for furan under their $NO_3$ + furan experimental conditions.*

115

*The α-terpinene $k(NO_3)$ determinations using 2,5-dimethylfuran and pyrrole as references yield results that are similar (but not identical) to that obtained using 2,3-dimethyl-2-butene as the reference, suggesting that they also probably do not react rapidly with $NO_2$. However, the observed differences between the systems could also result from the differences in $NO_2$ generation from the $NO_3$ + reference compound chemistry and the secondary effect on α-terpinene decay.*

120

We thank the reviewer for pointing out this potential interference in our experiments and providing the comment early to give us time to check these interferences experimentally.

125

In response to this comment we have conducted experiments looking at the $NO_2$ reaction with alpha-terpinene, furan, 2,5-dimethylfuran, and pyrrole. The experiments were performed with initial $NO_2$ mixing ratios of roughly 5 ppmv, similar to the maximum amount of $NO_2$ observed during the $NO_3$ experiments reported in the manuscript. The rate coefficient $k(\alpha\text{-terpinene} + NO_2)$ was determined by a fit to the measured $\alpha$-terpinene mixing ratio constrained by $[NO_2]$ (Figure 1 of response). The value obtained for the rate coefficient is $(5.8\pm0.8) \times 10^{-18}$ cm$^3$ molecule$^{-1}$ s$^{-1}$. This is in agreement with the value obtained by Atkinson et al. (1984) of $(6.5\pm1.4) \times 10^{-18}$ cm$^3$ molecule$^{-1}$ s$^{-1}$. For furan, 2,5-dimethylfuran, and pyrrole, the derived rate coefficients were all indistinguishable from zero under the experimental conditions employed, and can all be said to be $< 2.0 \times 10^{-20}$ cm$^3$ molecule$^{-1}$ s$^{-1}$. This is now discussed in the manuscript – see additional text below. However, the alpha-terpinene experiment is not reported, as these experiments have now been removed from the manuscript, see 'Major Changes' above.

130

135

[Figure]

**Figure 1** Experiment to determine $k(\alpha\text{-terpinene} + NO_2)$

140

*"A further potential interference with the current experimental setup, is the reaction of $NO_2$ with the compounds used. Rate coefficients have previously been measured for reaction of $NO_2$ with a number of unsaturated compounds here (Atkinson et al., 1984; Bernard et al., 2013). For conjugated dienes, these values can be large enough ($\sim 10^{-18}$ $cm^3$ molecule$^{-1}$ $s^{-1}$) to provide a significant loss under the experimental*

145

*conditions employed here. $NO_2$ is formed during these experiments from the decomposition of $N_2O_5$, with the $NO_2$ mixing ratio typically increasing up to roughly 3 ppmv through the experiment. Separate experiments were performed to look at the potential reaction of $NO_2$ with furan, 2,5-dimethylfuran and pyrrole. For all three compounds, their loss in the presence of $NO_2$ (allowing for dilution) was indistinguishable from zero, allowing an upper limit of $< 2 \times 10^{-20}$ $cm^3$ molecule$^{-1}$ $s^{-1}$ to be placed on their*

150

*$k(NO_2)$ rate coefficients."*

**Minor comments**

155 **Abstract: The common names furfural, α-angelicalactone and γ-crotonolactone are generally used for furan-2-aldehyde, 5-methyl-2(3H)-furanone and 2(5H)-furanone throughout the manuscript. It might be useful to include the common names in the abstract summary (i.e., as done on line 54).**

These names have been added in the abstract.

160

**Line 70: Please further clarify whether the $N_2O_5$ sample was introduced "continually" (i.e., repeatedly in aliquots) or "continuously" (i.e., without interruption in a constant flow throughout the experiment).**

The reviewer is correct, the sample was added continuously. This has been clarified in the text, which

165 now reads:

*"The reaction was then initiated by continuously introducing an $N_2O_5$ sample, held in a trap at $\sim$ 235 K with air flow of (2.5 – 5) L/min through it, for the duration of the experiment."*

170 **Lines 79-83: Cyclohexane (used as the reference for γ-crotonolactone) needs to be included in the materials list. Its rate coefficient should also be included in Table 2.**

This information has been added to the materials list and Table 2.

175 **Line 101: It would be helpful to have some explanation of why the rate of alkene and furan loss rates increase with time during the experiment (Fig. 1). It is not clear from the information given. At what point was the $N_2O_5$ flow added? It also would be good to have similar plots for other studied compounds in the Supplement.**

The VOCs were monitored for at least thirty minutes after addition, prior to the beginning of the $N_2O_5$

180 addition, to ascertain wall loss rates of the VOCs. The $N_2O_5$ addition was then continuous during the experiments. The trap slowly warms throughout the experiment, potentially increasing the rate of

addition of $N_2O_5$. When there are multiple VOCs in the chamber, as the experiment progresses, the $NO_3$ sink is reduced, as VOCs are removed. Therefore, even at a fixed rate of addition, the $NO_3$ concentration and hence the VOC loss rate of the slower reacting VOCs (cyclohexene in Figure 1) will noticeably increase through the experiment.

The SI now contains an additional six concentration-time plots, with the same setup as Figure 1 – see **Data Presentation / Availability** above.

*Line 113: Terminology: The term "rate(s)" is used incorrectly in many places where "rate coefficient(s)" should be used. As the authors are of course aware, a reaction rate (dimension: concentration/time) is the product of the rate coefficient and reagent concentration(s) – i.e., "k" is a rate coefficient, not a rate.*

This has been corrected throughout the manuscript, on lines 17, 18, 119, 120, 126, 127, 128, 143, 181, 203, 208, 211, 212, 227, 230, Table 4, 255, and 300.

*Line 138: Note that there is also a direct source of $HO_2$ from the reaction of $NO_3$ with α-terpinene. Like other conjugated cyclohexadienes, a minor H abstraction channel occurs to form a substituted cyclohexadienyl radical, followed by its reaction with $O_2$ to form p-cymene and $HO_2$ (Berndt et al., 1996).*
*Lines 134-150. Because the $NO_3$ + $HO_2$ reaction also forms OH, it might be worth mentioning that this is (presumably) uncompetitive under the experimental conditions.*

The following lines have been added to the paragraph to acknowledge this potential $HO_2$ source.

*"An additional minor source of $HO_2$ during the experiments will be H abstraction reactions by $NO_3$. These will produce $RO_2$ that can react to form RO radicals which may yield $HO_2$ following abstraction of an H atom by $O_2$. The rate coefficient of H abstraction by $NO_3$ is generally expected to be negligible relative to that of the $NO_3$ addition pathway. For $\alpha$-terpinene, Berndt et al. (1996) measured a yield of p-cymene, the aromatic product of the H abstraction pathway, of 6 %."*

*Line 206, Table 3: a-terpinene entries should be "α-terpinene".*

These have been changed.

*Line 215: The derived values of k($NO_3$) for 2,5-dimethylfuran and pyrrole are highlighted as being significantly smaller using α-terpinene as the reference. But, inspection of Table 3 suggests that the values obtained using 2,3-dimethyl-2-butene are further from the average in each case, the highest and lowest of the sets respectively.*

We agree that this was not made clear in the text / table. The k($NO_3$) values in Table 3 were calculated using the recommended value of k($\alpha$-terpinene+$NO_3$) from this manuscript. Using the k($\alpha$-terpinene+$NO_3$) value from the database (McGillen et al., 2020) gave the smaller values for 2,5-

dimethylfuran and pyrrole mentioned in the text. With the removal of the α-terpinene experiments, these lines have now been removed.

*Line 224. Replace "TME" by "2,3-dimethyl-2-butene".*

Replaced.

*Line 227, Table 4: "Kind et al. (2006)" or "Kind et al. (1996)"?*

The reviewer is correct, this should be Kind et al. (1996) throughout Table 4 and has been corrected.

*Line 347: Berndt et al. (1996) reference is missing.*

This reference has been added. In addition, on lines 151 and 270 the reference was given as Berndt et al. (1996) when it should have been Berndt et al. (1997). This has been changed.

**General comments**

*The manuscript presents a relative kinetic study in chamber simulation of a series of furans and relative compounds with $NO_3$ radical, major oxidant in the atmosphere by night. The studied compounds are furan, 2-methyl furan, 2,5-dimethylfuran, furan-2-aldehyde, 5-methyl-2(3H)-furanone, 2(5H)-furanone and pyrrole which are known to be emitted in the atmosphere during biomass burning.*

*This study reports for the first time rate coefficients for two furanones (α-angelicalactone and γ-crotonolactone) and investigate rate coefficients for the others compounds witch present few rate constant determinations in the literature. Although new kinetic studies are mandatory to complete and improve kinetic data bases the manuscript needs significant improvements before being published in ACP. The recommendations listed here after must absolutely be taken into consideration.*

**Major comments**

*1) As the paper includes a relative kinetic study I would expect a detailed presentation of the experimental conditions, analysis of data, results and discussion. However, there is a certain number of information missing from the manuscript (and/or SI) to allow a proper evaluation of the quality of the data. My main comment is that relative rate plots and time series of concentrations are missing for the majority of the compounds. The plots are needed for example to investigate the presence of an eventual secondary chemistry or products interference. This is of particular interest for the compounds for which the rate constants obtained are not in agreement with the literature.*

*- please present relative rate plots and time series of concentrations for compounds of interest (VOC, reference compound and $NO_2$) for 2,5-dimethyl furan, pyrrole, furfural, γ-crotonolactone, α-terpinene;*
*- please present relative rate plots and time series of concentrations for reference compound and $NO_2$ for α-angelicalactone;*
*Although there is no need to include all the time series and plots in the manuscript I suggest to complete the manuscript by including time series of concentration and relative rate plots for compounds for which this is the first rate constant determination (α-angelicalactone) or/and those experiencing fast reactivity (e.g 2,5-dimethylfuran or pyrrole). Representative plots that are not shown in the manuscript must be included in SI for all compounds.*

Figure 2 is now a six panel plot containing relative rate plots for all experiments for all six compounds (furan, 2-methylfuran, 2,5-dimethylfuran, furfural, pyrrole, $\alpha$-angelicalactone).

The SI now contains an additional six example concentration-time plots, with the same setup as Figure 1.

All of the data (raw output from the FTIR, and concentration-time data) will be made available at https://data.eurochamp.org with a doi.

*2) The technique used for monitor VOCs is an in situ IRTF.*
*- The absorption bands used for reference compounds are missing. Please add a table with the missing information.*

These have been added to Table 1.

*- The technique is not selective and reactants as well as the products of reaction are expected to absorb IR light and be presented on the absorption spectra at the level of concentration used here. Some of the products have absorption bands that may interfere with the reactant bands and thus perturb the data analysis. Is this the case? Can you comment on that?*

The reviewer is right that products may have absorption bands that overlap with those being used to follow the reactants. This is generally not a problem for analysis if the reactant bands are either sharp peaks, as is the case for most of the furans, or if there is only a partial overlap. It is only really if the reactant peaks are broad and the product peaks have a very similar profile, that analysis becomes more challenging. As presented in Table 1, all of the compounds used have a number of possible absorptions which could be used if there is an interference with either a reference, or with a product of the compound or reference. If a compound had multiple absorption features, all were checked for the analysis even if the primary absorption feature appeared to have no interference.

*- Generation of NO$_3$ via the thermal dissociation of N$_2$O$_5$ implies the presence of N$_2$O$_5$, NO$_2$ and HNO$_3$ (depending of the purity of the N$_2$O$_5$ synthesis) in the chamber. IRTF allows also to monitor these species. Please add information about the levels of these species in the experiments.*

The following text has been added to Section 2.4 (Analysis). NO$_2$ concentrations are now also discussed later in Section 2.4 concerning its potential interference with the calculated relative rate coefficients.

*"N$_2$O$_5$ was not present at detectable levels (by FTIR) during most of the experiments. The only experiments in which N$_2$O$_5$ concentrations built up in the chamber, were those with the slowest reacting VOCs, i.e. furfural and $\gamma$-crotonolactone. NO$_2$ concentrations increased throughout all experiments, typically up to 2 − 3 ppmv. The NO$_2$ is initially produced from the decomposition of N$_2$O$_5$, and later potentially by the loss of NO$_2$ from nitrated VOCs / nitrated radicals. HNO$_3$ concentrations increased throughout the experiments, typically up to 3 − 4 ppmv. This could be either due to impurities in the N$_2$O$_5$ sample, or from H abstraction reactions of NO$_3$. It is not thought that this level of HNO$_3$ will cause any interference in the rate coefficient determinations."*

*3) Although α-terpinene is a compound of interest in this work, there is very few information regarding this compound in the manuscript:*
*- The title and introduction of the paper are focused mainly on furans*
*- No information regarding the data analysis is presented (absorption bands in Table 1, absorption spectra in SI). Please complete*
*- The compound is used as reference compound in two experiments (2,5-dimethylfuran, pyrrole) but the recommended rate and uncertainties are not present in Table 2. Please complete.*
*- No concentration profiles, no relative rate plots are presented. Please complete.*
*- Not included in Figure 3 neither in table S1. Please complete.*
*- No discussion or explanation is given on the faster rate constant regarding the previous rate constant determinations. I would expect further discussion as previous values are in good agreement within the uncertainties.*

The experiments using $\alpha$-terpinene as a reference have now been removed from the manuscript, with justification as given in the opening statement above.

It is noted that we mistakenly presented the literature rate coefficients in Table 4 from Fouqueau et al. (2020), where they were given relative to $k$(2,3-dimethyl-2-butene+NO$_3$) = 5.5 $\times10^{-11}$ cm$^3$ molecule$^{-1}$ s$^{-1}$, rather than 5.7 $\times10^{-11}$ cm$^3$ molecule$^{-1}$ s$^{-1}$, the recommendation from McGillen et al. (2020) used elsewhere in our manuscript.

It is also noted that, as discussed in Fouqueau et al. (2020), the two previous relative rate determinations (Atkinson et al., 1985 and Berndt et al., 1996) are not in good agreement. The recommended rate coefficients normalised to $k$(2,3-dimethyl-2-butene + NO$_3$) = 5.7$\times10^{-11}$ cm$^3$ molecule$^{-1}$ s$^{-1}$, are (1.81±0.47)$\times10^{-10}$ cm$^3$ molecule$^{-1}$ s$^{-1}$ (Atkinson et al. 1985) and (1.02±0.06)$\times10^{-10}$ cm$^3$ molecule$^{-1}$ s$^{-1}$ (Berndt et al., 1996). And much of the uncertainty attributed to the rate coefficient of Atkinson et al. is related to the uncertainty of the rate coefficient of the reference. When assessing the values and stated uncertainties for the relative rate coefficient $k$($\alpha$-terpinene+NO$_3$)/$k$(2,3-dimethyl-2-butene+NO$_3$): Atkinson (3.18±0.13), Berndt (1.796±0.1), it becomes clear that there is a large discrepancy between these measurements. The value of (1.2±0.3)$\times10^{-10}$ cm$^3$ molecule$^{-1}$ s$^{-1}$ determined by Fouqueau et al. (2020) using an absolute method, lies in between the two relative rate measurements.

*-α-terpinene was used with 90% of purity without further purification. Can you specify the nature of the impurity and discuss impact on the rate constant?*

The $\alpha$-terpinene sample was used as provided by Sigma-Aldrich. Stated impurities are 1,4-cineole and 1,8-cineole. As discussed above, in the FTIR each compound has a unique absorption signature. So these impurities would only be a potential problem if they were to have very similar absorption features to $\alpha$-terpinene.

Furthermore, if cineole was causing an interference, it a reported k(NO$_3$) of 1.7e-16, i.e. roughly six orders of magnitude smaller than $\alpha$-terpinene. As such, a significant interference would make the derived rate coefficient smaller than expected, rather than larger as observed in our experiments.

365

*In my opinion the data regarding this compound should be further presented and discuss or the authors should take in consideration to remove it from the manuscript.*

370

*4) While authors have investigated a possible OH generation and thus reactions of VOC of interest with OH no other sinks in the experiments were considered or discuss. The generation of NO$_3$ radicals by thermal dissociation of N$_2$O$_5$ has indeed the advantage to work in a O$_3$ free environment but implies large amounts*

375

*of NO$_2$. I agree with the first referee, NO$_2$ may be possible sink for the studied compounds in the experimental conditions presented here (large concentrations of VOC, and N$_2$O$_5$). Authors should investigate this possible path for all compounds but especially for rate constants found higher that the already published ones (pyrrole, 2,5-dimethylfuran and $\alpha$-terpinene).*

380

Experiments have been performed to examine NO$_2$ reaction with $\alpha$-terpinene, furan, 2,5-dimethyl furan, and pyrrole in response to reviewer 1's comments. It was concluded that the reaction rate coefficient is < $2.0 \times 10^{-20}$ cm$^3$ molecule$^{-1}$ s$^{-1}$ for the latter three compounds. Please see the discussion above for further details.

385 **Minor comments**

*Furan -2-aldehyde, 5 methyl-2(3H)-furanone and 2(H)-furanone are used either by their scientific names (eg. abstract, table 1) or by their common names (furfural, $\alpha$-angelicalactone, $\gamma$-crotonolactone) ( eg. Materiels, Table 3) which is difficult to follow. For clarity, please homogenize.*

390

The common names are now used throughout the manuscript. When mentioned in the abstract and in the introduction, the scientific name is given in brackets after the common name.

*Line 43: The sentence "Furan type compounds are removed from the atmosphere by reaction with the major oxidants OH, NO$_3$ and O$_3$" should be introduce earlier (line 34) for clarity.*

395

The introduction is ordered by first, a discussion of sources of furans to the atmosphere, and second a discussion of their sinks. There is admittedly a brief discussion of furan oxidation prior to discussion of the oxidants, the purpose of which is to introduce the source of 2-furanones (i.e. production in the atmosphere from furan oxidation).

400

**Experimental**

*More information about the experiments is needed in my opinion.*

*Line 62: The authors mention that the experiments were conducted with the chamber operated at a slight overpressure to compensate from" removal of air sampling and to prevent ingress of outside air to the chamber". The reason for "air sampling" is unclear as the only instrument mentioned for these experiments is an in situ IRTF. Please clarify.*

It is true that removal of air through sampling was minimal. An ozone monitor was running during the experiments which will remove a very small amount of air.

*Line 67: Specify the order of introduction of VOCs (VOC of interest followed by Reference VOC?), continuous injection of $N_2O_5$, …*

There was no set order of introduction of VOCs. If a reference spectrum was required for one of the VOCs then this was added first and the reference spectrum taken. The VOCs were then monitored for at least half an hour prior to the beginning of the continuous addition of $N_2O_5$ (see discussion on this point in answer to review 1 above) to ascertain wall loss rates.

*Line 74: Please specify the time and spectral resolution for the IR spectra*

The spectral resolution was 0.25 cm$^{-1}$. As stated, each scan consisted of 30 or 60 co-additions. These took a time of 2 or 4 minutes. The text has been amended accordingly and now reads:

*"Each scan was comprised of either 30 or 60 co-additions, taking a total of 2 or 4 minutes respectively, depending on the expected rate of loss of the VOCs, with a spectral resolution of 0.25 cm$^{-1}$."*

**Materials**

*For clarity, please differentiate interest VOC from reference VOC.*

This has been added.

**Results and discussion**

*Table 3: A number of experiments are missing. Please complete.*

It is not clear to what the reviewer is referring in Table 3. We note that the number of repeats was missing for some of the compounds in Table 3 and have now completed this, we assume that this is to what the reviewer refers.

445 ***Table 4: For α-terpinene: information are missing regarding previous studies: i) for Atkinson et al., et Berndt et al. please specify the type of study. For Fouqueau et al., 2020 please specify the study and method used.***

The α-terpinene information has now been removed.

450

**SI**

***Table S1: for clarity: replace "NA" by "-"; reorganize the table (e.g. by Compound 1 or Reference)***

This has been corrected and the table has been ordered by Compound 1.

455

---

## Author Response (AR2)

**Response to Reviewers**

15/12/21

**Response to Reviewers of:**

**NO$_3$ chemistry of wildfire emissions: a kinetic study of the gas-phase reactions of furans with the NO$_3$ radical**
**by Newland et al., 2021, submitted to ACP**

**General Response**

We thank the reviewers for giving up more of their time to make further insightful comments, helping to clarify and improve our manuscript. Responses to each reviewer are given below. Responses to specific points raised by each reviewer are given separately beneath that point. Reviewers' comments are bold and italic, the authors' comments are inset in plain type.

**Reviewer 1**

**This revised manuscript reports a relative rate study of the reactions of NO3 with a series of furans and related compounds, which are known to be important components of biomass burning emissions. The authors have taken account of the reviewers' comments on the original version, leading to improvements in the analysis and manuscript.**

**Importantly, the authors have considered the potential reaction of NO2 with the target and reference compounds, which was found to be significant for the reference compound a-terpinene. The experiments using a-terpinene as a reference have therefore logically been excluded from the revised analysis. As a result of this and other improvements, the paper is now acceptable for publication in ACP. I have a few minor/technical corrections and suggestions, which are listed below.**

**Following removal of the a-terpinene experiments, a few rogue pieces of information appear not to have been deleted:**

**Line 82: "a-terpinene" is still listed in the Materials section.**

> *Done*

**Line 181: The sentence "A recommendation of an updated rate coefficient for a-terpinene+NO3 is also given in Table 3." needs to be deleted.**

> *Done*

**The Berndt et al. (1996) and Fouqueau et al. (2020) references no longer appear to be required.**

> *Done*

**Other comments:**

**Line 60: The Zhou et al. (2017) reference is missing. I could not check experimental details**

elsewhere (e.g. if the chamber is fixed volume or collapsible) - hence my query about dilution below.

*This reference has been added to the reference list*

**Line 69: Presumably the SF6 is added to monitor dilution when the N2O5/air mixture is continuously added during the experiment. It is therefore not clear why there should be any dilution during the 30 minute standing time, unless a dummy air flow is included. Does the specified dilution rate (line 70) refer to the experiment or the 30 minutes standing time?**

*A flow of 5 L/min of purified air was continuously added through the experiment (this is the same flow that is used to add the samples and $N_2O_5$). Air is then also removed to maintain a constant pressure (at a slight overpressure to prevent possible ingress of air from outside the chamber). Hence the dilution rate should remain the same throughout the whole experiment period (i.e. both prior to and after addition of N2O5). The following sentence has been added to the experimental approach to clarify this:*

> "A flow of 5 L/min of purified air was continuously added throughout the experiment, and air is then removed from the chamber to maintain a constant pressure (this is a slight overpressure to prevent possible ingress of air from outside the chamber)."

**Line 171: The authors' discussion comment could be referred to here for further details on the NO2 experiments.**

*We have added the following line from the discussion comment to provide more detail on the $NO_2$ experiments:*

> "The experiments were performed with initial VOC mixing ratios of 3 ppmv, and initial $NO_2$ mixing ratios of roughly 5 ppmv, similar to the maximum amount of $NO_2$ observed during the $NO_3$ experiments."

**Line 199: "....the values for furan and 2,5-dimethylfuran are ~ 50 % and 100 % greater respectively." To make the statement a bit clearer, I suggest inserting "reported here" after "2,5-dimethylfuran".**

*Done*

**Line 217: This sentence appears to say that cyclohexane was selectively diluted by 10 %, which is clearly impossible. Does this mean that a 10% loss of g-crotonolactone could not be measured reliably?**

*The sentence is supposed to suggest that 10 % of the cyclohexane was removed by reaction with $NO_3$. I.e. both compounds will have had a certain loss to dilution, but cyclohexane has an additional chemical loss to $NO_3$ on top of that, whereas g-crotonolactone does not. The sentence has been changed to the following to hopefully make it slightly more clear:*

"In this experiment, roughly 10 % of the cyclohexane was removed by reaction with NO₃ (accounting for loss by dilution), whereas there was no appreciable chemical loss of γ-crotonolactone."

**Line 230 (Table 3): Should the 2,5-dimethylfuran entries with pyrrole and 2-methylfuran as references have E-11 exponents?**

*Yes, thank you! This has been changed.*

**Line 246: The comparison in Table 5 is a very good way of summarizing the comparative reactivity of the compounds to the different oxidants, and should be included. However, I am always a little nervous of presenting such information as "black and white" conclusions. The oxidant concentrations are very variable and, particularly for NO3, are very conditions dependent. If there is any residual NO in a plume, NO3 is removed very quickly and its concentration is suppressed; and if NO is completely removed, there must be enough O3 remaining to allow NO3 to be formed from O3 + NO2. Also, it is noted in the second sentence of the abstract, that the (presumably elevated) concentrations of the target compounds results in a likely elevated NO3 reactivity under biomass plume conditions, which might mean that [NO3] can easily be suppressed relative to "typical" night-time or daytime levels when the NO3 reactivity is lower. Perhaps a little more qualification in the discussion is required.**

*The following sentence has been added in the Atmospheric Implications section, at line 268:*

"It is noted that oxidant concentrations have a high spatial and temporal variability due to variability in their sources and sinks, and that oxidant levels within biomass burning plumes in particular are poorly understood. Hence the relative importance of the oxidants shown in Table 5 is likely to vary dependent on conditions."
* * *
**Reviewer 2**

**The authors have properly addressed my comments as well as those of the other review. I find the new manuscript harmonized, clear and complete.**
**I still have some minor comments that need to be addressed by the authors before publication. The comments are listed below.**

**Line 82 : Please remove "α-terpinene (90%, Sigma-Aldrich)"**

*Done*

**Line 171: Please add text to justify why the rate coefficients for reaction of NO2 with 2-methylfuran, furfural, and α-angelicalactone have not been investigated.**

*We have added the following sentence after line 171:*

> "Based on these experiments, it was assumed that the $k(NO_2)$ rate coefficients for 2-methylfuran, furfural, and a-angelicalactone are likely to be of a similar magnitude, and hence provide negligible interference under the experimental conditions employed."

**Line 182: please remove: "A recommendation of an updated rate coefficient for a-terpinene+NO3 is also given in Table 3".**

*Done*

**Table 3: A number of experiment repeats are still missing for:**
**- α-angelicalactone with α-pinene and cyclohenxane as reference compounds. Please complete.**
**- Furan with cyclohexane, α-pinene and camphene as reference compounds. Please complete.**
**- γ-crotonolactone with cyclohexane as reference compound. Please complete.**

*Done*

**- For furfural, it seems like the value of the rate constant kNO3 with furan as reference compound is missing from the table while the experiment is plotted in Figure 2. Please complete.**

*Done*